# On Memorization in Probabilistic Deep Generative Models

**Gerrit J.J. van den Burg**[*]
gertjanvandenburg@gmail.com

**Christopher K.I. Williams**
University of Edinburgh
The Alan Turing Institute
ckiw@inf.ed.ac.uk

## Abstract

Recent advances in deep generative models have led to impressive results in a variety of application domains. Motivated by the possibility that deep learning models might memorize part of the input data, there have been increased efforts to understand how memorization arises. In this work, we extend a recently proposed measure of memorization for supervised learning (Feldman, 2019) to the unsupervised density estimation problem and adapt it to be more computationally efficient. Next, we present a study that demonstrates how memorization can occur in probabilistic deep generative models such as variational autoencoders. This reveals that the form of memorization to which these models are susceptible differs fundamentally from mode collapse and overfitting. Furthermore, we show that the proposed memorization score measures a phenomenon that is not captured by commonly-used nearest neighbor tests. Finally, we discuss several strategies that can be used to limit memorization in practice. Our work thus provides a framework for understanding problematic memorization in probabilistic generative models.

## 1 Introduction

In the last few years there have been incredible successes in generative modeling through the development of deep learning techniques such as variational autoencoders (VAEs) [1, 2], generative adversarial networks (GANs) [3], normalizing flows [4, 5], and diffusion networks [6, 7], among others. The goal of generative modeling is to learn the data distribution of a given data set, which has numerous applications such as creating realistic synthetic data, correcting data corruption, and detecting anomalies. Novel architectures for generative modeling are typically evaluated on how well a complex, high dimensional data distribution can be learned by the model and how realistic the samples from the model are. An important question in the evaluation of generative models is to what extent training observations are *memorized* by the learning algorithm, as this has implications for data privacy, model stability, and generalization performance. For example, in a medical setting it is highly desirable to know if a synthetic data model could produce near duplicates of the training data.

A common technique to assess memorization in deep generative models is to take samples from the model and compare these to their nearest neighbors in the training set. There are several problems with this approach. First, it has been well established that when using the Euclidean metric this test can be easily fooled by taking an image from the training set and shifting it by a few pixels [8]. For this reason, nearest neighbors in the feature space of a secondary model are sometimes used, as well as cropping and/or downsampling before identifying nearest neighbors (e.g., [9–11]). Second, while there may not be any neighbors in the training set for a small selection of samples from the model, this does not demonstrate that there are *no* observations that are highly memorized. Indeed, in several

---

[*]Work done while at The Alan Turing Institute.

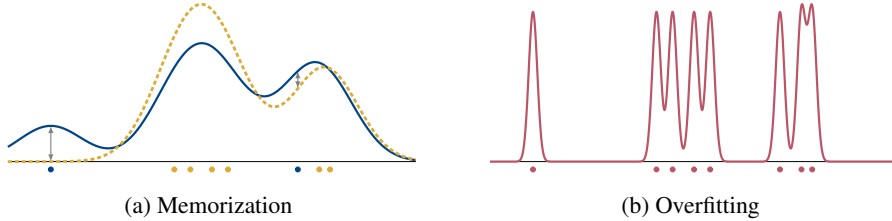

(a) Memorization                  (b) Overfitting

Figure 1: Memorization and overfitting. In (a) the solid blue curve reflects the probability density when all observations are included, whereas the dashed yellow curve is the density when only yellow observations are included. The local change in density that occurs when an observation is removed from the training data indicates the extent to which the model memorized the observation. The density typically associated with overfitting due to overtraining is shown by the solid red curve in (b).

recent publications on deep generative models it is possible to identify observations highly similar to the training set in the illustrations of generated samples (see Supplement A).

Memorization in generative models is not always surprising. When the training data set contains a number of highly similar observations, such as duplicates, then it would be expected that these receive an increased weight in the model and are more likely to be generated. The fact that commonly-used data sets contain numerous (near) duplicates [12] therefore provides one reason for memorization of training observations. While important, memorization due to duplicates is not the focus of this work. Instead, we are concerned with memorization that arises as *an increased probability of generating a sample that closely resembles the training data in regions of the input space where the algorithm has not seen sufficient observations to enable generalization*. For example, we may expect that highly memorized observations are either in some way atypical or are essential for properly modeling a particular region of the data manifold.

Figure 1a illustrates this kind of local memorization in probabilistic generative models. We focus on explicit density models as these are more amenable to a direct analysis of the learned probability distribution (as opposed to *implicit* density models such as GANs). The figure shows that in certain regions of the input space the learned probability density can be entirely supported by a single, potentially outlying, observation. When sampling from the model in these parts of the space it is thus highly likely that a sample similar to an input observation will be generated. The figure also illustrates that in regions of the input space that are densely supported by closely-related observations, sampling will yield observations that resemble the input data. The change in the probability density of an observation that occurs when it is removed from the data forms the basis of the memorization score we propose in Section 3. This form of memorization should be contrasted with what is commonly associated with memorization due to overfitting (illustrated in Figure 1b). Overfitting is a *global* property of a model that typically occurs when it is trained for too long or with too high a learning rate (i.e., *overtraining*), so that a gap develops between the training and test performance. Thus, we emphasize that in generative models memorization and generalization can occur simultaneously at distinct regions of the input space, and that memorization is not necessarily caused by overtraining.

To understand memorization further, consider the simple case of fitting a multivariate normal distribution. In this scenario, the presence or absence of a particular observation in the data set will have a small effect on the learned model unless the observation is an outlier. By contrast, a kernel density estimate (KDE) [13, 14] of the probability density may be more sensitive to the presence or absence of a particular observation. To see why this is the case, consider that in sparsely-populated regions of the input space the KDE can be supported by a relatively small number of observations. Although deep generative models typically operate in much higher dimensional spaces than the aforementioned methods, the same problem can arise when generalizing to regions of the space that are weakly supported by the available data. Because these models are optimized globally, the model has to place some probability mass in these regions. As we will demonstrate below, it is not necessarily the case that the model places low probability on such observations, resulting in observations that are both highly memorized and not significantly less likely under the model than other observations.

In this work, we extend a recently proposed measure of memorization for supervised learning [15, 16] to probabilistic generative models and introduce a practical estimator of this memorization score. We subsequently investigate memorization experimentally, where we focus on the variational autoencoder.

In our experiments we demonstrate that highly memorized observations are not necessarily outliers, that memorization can occur early during the training process, and show the connection between nearest neighbor tests for memorization and the proposed memorization score. Finally, we discuss approaches that can limit memorization in practice.

## 2   Related Work

Here we review work on memorization in deep learning, memorization as it relates to membership inference, the evaluation of generative models, as well as influence functions and stability.

**Memorization in deep learning.**   The observation that deep learning models can learn from patterns of random data has been a catalyst for recent efforts to understand memorization in supervised learning [17–19]. A number of approaches have been proposed to test for memorization in specific applications. In [20] memorization in language models is evaluated using a "canary" string (e.g., if "my social security number is $x$" is in the training data, how often does the model complete the prompt "my social security number is" using $x$ instead of a comparable $y \neq x$ that is not in the training set). Unfortunately, this approach does not translate easily to other contexts, such as images. Moreover, language models often contain explicit memory cells such as LSTMs [21] that can facilitate memorization, which are absent in most generative models.

A memorization score for supervised learning was proposed in [15], which forms the inspiration for our formulation in Section 3. A related "consistency score" for supervised learning was proposed in [22]. We argue, however, that memorization in supervised learning differs fundamentally from that in generative models, as the label prediction task affects the training dynamics, and label noise is known to induce memorization in supervised learning [23, 24]. Building on earlier work by [25, 26], a hypothesis test is proposed in [27] that is based on the premise that memorization has occurred when samples from the trained model are "closer" to the training data than observations from the test set. While this is a useful test for aggregate memorization behavior in (a region of) the input space, our proposed score function allows us to quantify the memorization of a single observation.

**Membership inference.**   A topic closely related to memorization is the problem of membership inference. Here, the goal is to recover whether a particular observation was part of the unknown training data set, either using knowledge of the model, access to the model, or in a black-box setting. Membership inference is particularly important when models are deployed [28], as potentially private data could be exposed. In the supervised learning setting, [29] propose to use an attack model that learns to classify whether a given sample was in the training set. Later work [30, 31] focused on generative models and proposed to train a GAN on samples from the target model. The associated discriminator is subsequently used to classify membership of the unknown training set. A related approach to recovering training images is described in [32], using an optimization algorithm that identifies for every observation the closest sample that can be generated by the network. However this requires solving a highly non-convex problem, which isn't guaranteed to find the optimal solution.

**Evaluating generative models.**   Memorization is a known issue when evaluating generative models, in particular for GANs [26, 33]. Several approaches are discussed in [8], with a focus on the pitfalls of relying on log-likelihood, sample quality, and nearest neighbors. Using the log-likelihood can be particularly problematic as it has been shown that models can assign higher likelihood to observations outside the input domain [34]. Nowadays, generative models are frequently evaluated by the quality of their samples as evaluated by other models, as is done in the Inception Score (IS) [35] and Fréchet Inception Distance (FID) [36]. Since these metrics have no concept of where the samples originate, the pathological case where a model memorizes the entire training data set will yield a near-perfect score. Motivated by this observation, [37] propose to use neural network divergences to measure sample diversity and quality simultaneously, but this requires training a separate evaluation model and there is no guarantee that *local* memorization will be detected.

**Influence & Stability.**   The problem of memorization is also related to the concept of *influence functions* in statistics [38, 39]. Influence functions can be used to measure the effect of upweighting or perturbing an observation and have recently been considered as a diagnostic tool for deep learning [40, 41]. However, it has also been demonstrated that influence function estimates in deep learning models can be fragile [42]. Below, we therefore focus on a relatively simple estimator to gain a

reliable understanding of memorization in probabilistic deep generative models. Concurrent work in [43] focuses on influence functions for variational autoencoders based on the "TracIn" approximation of [44] computed over the training run. An important difference is that while the method of [43] is computed for one particular model, our memorization score applies to a particular model *architecture* by averaging over multiple $K$-fold cross-validation fits (see Section 3). Related to influence functions is the concept of stability in learning theory [45]. In particular, the *point-wise hypothesis stability* is an upper bound on the expected absolute change in the loss function when an observation is removed from the training set (where the expectation is over all training sets of a given size). We instead focus on the change in the density of a probabilistic model when trained on a specific data set.

## 3   Memorization Score

We present a principled formulation of a memorization score for probabilistic generative models, inspired by the one proposed recently in [15, 16] for supervised learning. Let $\mathcal{A}$ denote a randomized learning algorithm, and let $a$ be an instance of the algorithm (i.e., a trained model). Here, $\mathcal{A}$ captures a complete description of the algorithm, including the chosen hyperparameters, training epochs, and optimization method. The randomness in $\mathcal{A}$ arises from the particular initial conditions, the selection of mini batches during training, as well as other factors. Denote the training data set by $\mathcal{D} = \{\mathbf{x}_i\}_{i=1}^n$ with observations from $\mathcal{X} \subseteq \mathbb{R}^D$. Let $[n] = \{1, \ldots, n\}$ and write $\mathcal{D}_\mathcal{I} = \{\mathbf{x}_i : \mathbf{x}_i \in \mathcal{D}, i \in \mathcal{I}\}$ for the subset of observations in the training data indexed by the set $\mathcal{I} \subseteq [n]$. The posterior probability assigned to an observation $\mathbf{x} \in \mathcal{X}$ by a model $a$ when trained on a data set $\mathcal{D}$ is written as $p(\mathbf{x} \,|\, \mathcal{D}, a)$.

We are interested in the posterior probability of an observation assigned by the algorithm $\mathcal{A}$, not merely by an instantiation of the algorithm. Therefore we introduce the probability $P_\mathcal{A}(\mathbf{x} \,|\, \mathcal{D})$ and its sampling estimate as

$$P_\mathcal{A}(\mathbf{x} \,|\, \mathcal{D}) = \int p(\mathbf{x} \,|\, \mathcal{D}, a) p(a) \,\mathrm{d}a \approx \frac{1}{T} \sum_{t=1}^T p(\mathbf{x} \,|\, \mathcal{D}, a_t), \tag{1}$$

for some number of repetitions $T$. We see that $P_\mathcal{A}(\mathbf{x} \,|\, \mathcal{D})$ is the expectation of $p(\mathbf{x} \,|\, \mathcal{D}, a)$ over instances of the randomized algorithm $\mathcal{A}$.

To facilitate meaningful interpretation of the memorization score we use the difference in log probabilities, in contrast to [15, 16]. Thus we define the leave-one-out (LOO) memorization score as

$$M^{\mathrm{LOO}}(\mathcal{A}, \mathcal{D}, i) = \log P_\mathcal{A}(\mathbf{x}_i \,|\, \mathcal{D}) - \log P_\mathcal{A}(\mathbf{x}_i \,|\, \mathcal{D}_{[n] \setminus \{i\}}). \tag{2}$$

This memorization score measures how much more likely an observation is when it is included in the training set compared to when it is not. For example, if $M^{\mathrm{LOO}}(\mathcal{A}, \mathcal{D}, i) = 10$, then $P_\mathcal{A}(\mathbf{x}_i \,|\, \mathcal{D}) = \exp(10) \cdot P_\mathcal{A}(\mathbf{x}_i \,|\, \mathcal{D}_{[n] \setminus \{i\}})$. Moreover, when $M^{\mathrm{LOO}}(\mathcal{A}, \mathcal{D}, i) = 0$ removing the observation from the training data has no effect at $\mathbf{x}_i$, and when $M^{\mathrm{LOO}}(\mathcal{A}, \mathcal{D}, i) < 0$ the observation is more likely under the model when it is removed from the training data. We will abbreviate the LOO memorization score as $M_i^{\mathrm{LOO}} := M^{\mathrm{LOO}}(\mathcal{A}, \mathcal{D}, i)$ when the arguments are clear from context.

**Estimation.**   The memorization score in (2) is a leave-one-out estimator that requires fitting the learning algorithm $\mathcal{A}$ multiple times for each observation as it is left out of the training data set. As this is computationally infeasible in general, we introduce a practical estimator that simplifies the one proposed in (2). Instead of using a leave-one-out method or random sampling, we use a $K$-fold approach as is done in cross-validation. Let $\mathcal{I}_k$ denote randomly sampled disjoint subsets of the indices $[n] = \{1, \ldots, n\}$ of size $n/K$, such that $\cup_{k=1}^K \mathcal{I}_k = [n]$. We then train the model on each of the training sets $\mathcal{D}_{[n] \setminus \mathcal{I}_k}$ and compute the log probability for all observations in the training set and the holdout set $\mathcal{D}_{\mathcal{I}_k}$.

Since there is randomness in the algorithm $\mathcal{A}$ and in the chosen folds $\mathcal{I}_k$, we repeat the cross-validation procedure $L$ times and average the results. Writing $\mathcal{I}_{\ell,k}$ for the $k$-th holdout index set in run $\ell$ and

---

**Algorithm 1** Computing the Cross-Validated Memorization Score

---

**Input:** Algorithm $\mathcal{A}$, data set $\mathcal{D}$, repetitions $L$, folds $K$
**Output:** $M_i^{\text{K-fold}}, \forall i$
1: **for** $\ell = 1, \ldots, L$ **do**
2:      $\mathcal{G}_\ell \leftarrow$ Random partition of $[n]$ into $K$ disjoint subsets
3:      **for** $\mathcal{I}_{\ell,k} \in \mathcal{G}_\ell$ with $k = 1, \ldots, K$ **do**
4:          $a_{\ell,k} \leftarrow$ Train $\mathcal{A}$ on $\mathcal{D}_{[n] \setminus \mathcal{I}_{\ell,k}}$
5:          $\pi_{\ell,k,i} \leftarrow$ Compute $\log p(\mathbf{x}_i \,|\, \mathcal{D}_{[n] \setminus \mathcal{I}_{\ell,k}}, a_{\ell,k}), \forall i \in [n]$
6:      **end for**
7: **end for**                   $\triangleright$ LogMeanExp$(\{u_i\}_{i=1}^n) = -\log n +$ LogSumExp$(\{u_i\}_{i=1}^n)$
8: $U_i \leftarrow$ LogMeanExp$(\{\pi_{\ell,k,i} : \ell \in [L], k \in [K], i \notin \mathcal{I}_{\ell,k}\}), \forall i$
9: $V_i \leftarrow$ LogMeanExp$(\{\pi_{\ell,k,i} : \ell \in [L], k \in [K], i \in \mathcal{I}_{\ell,k}\}), \forall i$
10: $M_i^{\text{K-fold}} \leftarrow U_i - V_i, \forall i$

---

abbreviating the respective training set as $\mathcal{D}_{\ell,k} = \mathcal{D}_{[n] \setminus \mathcal{I}_{\ell,k}}$, the memorization score becomes

$$M_i^{\text{K-fold}} = \log\left[ \frac{1}{L(K-1)} \sum_{\ell=1}^{L} \sum_{k=1}^{K} \mathbb{1}_{i \notin \mathcal{I}_{\ell,k}} p(\mathbf{x}_i \,|\, \mathcal{D}_{\ell,k}, a_{\ell,k}) \right] \tag{3}$$

$$- \log\left[ \frac{1}{L} \sum_{\ell=1}^{L} \sum_{k=1}^{K} \mathbb{1}_{i \in \mathcal{I}_{\ell,k}} p(\mathbf{x}_i \,|\, \mathcal{D}_{\ell,k}, a_{\ell,k}) \right],$$

where $\mathbb{1}_v$ is the indicator function that equals 1 if $v$ is true and 0 otherwise. Each of the $K-1$ folds where observation $i$ is in the training set contributes to the first term in (2), and when observation $i$ is in the holdout set it then contributes to the second term. This approach is summarized in Algorithm 1, where log probabilities are used for numerical accuracy. In practice, the number of repetitions $L$ and folds $K$ will be dominated by the available computational resources.

**When is memorization significant?** A natural question is what values of the memorization score are significant and of potential concern. The memorization scores can be directly compared between different algorithm settings on the same data set, for instance to understand whether changes in hyperparameters or model architectures increase or decrease memorization. Statistical measures such as the mean, median, and skewness of the memorization score or the location of, say, the 95th percentile, can be informative when quantifying memorization of a particular model on a particular data set, but can not necessarily be compared between data sets. In practice, we also find that the distribution of the memorization score can differ between modes in the data set, such as distinct object classes. This can be understood by considering that the variability of observations of distinct classes likely differs, which affects the likelihood of the objects under the model, and in turn the memorization score. We will return to this question in Section 6.

## 4 Experiments

We next describe several experiments that advance our understanding of memorization in probabilistic deep generative models, with a focus on the variational autoencoder setting. Additional results are available in Supplement C. Code to reproduce our experiments can be found in an online repository.[2]

### 4.1 Background

We employ the variational autoencoder (VAE) [1, 2] as the probabilistic generative model in our experiments, although it is important to emphasize that the memorization score introduced above is equally applicable to methods such as normalizing flows, diffusion networks, and other generative models that learn a probability density over the input space. The VAE is a latent-variable model, where we model the joint distribution $p_\theta(\mathbf{x}, \mathbf{z})$ of an observation $\mathbf{x} \in \mathcal{X} \subseteq \mathbb{R}^D$ and a latent variable $\mathbf{z} \in \mathcal{Z} \subseteq \mathbb{R}^d$. The joint distribution can be factorized as $p_\theta(\mathbf{x}, \mathbf{z}) = p_\theta(\mathbf{x} \,|\, \mathbf{z})p(\mathbf{z})$, and in the VAE the prior distribution $p(\mathbf{z})$ is typically assumed to be a standard multivariate Gaussian. The posterior

---

[2]See: https://github.com/alan-turing-institute/memorization.

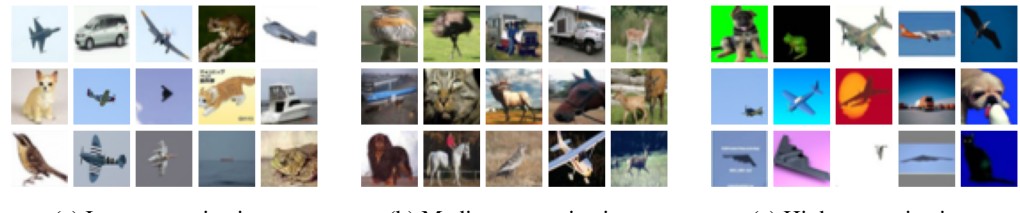

|                          |                             |                          |
|:------------------------:|:---------------------------:|:------------------------:|
| (a) Low memorization     | (b) Median memorization     | (c) High memorization    |

Figure 2: Observations with low, median, and high memorization scores in the CIFAR-10 data set, when learning the distribution with a convolutional VAE. Memorization scores range from about $-180$ in the top left of figure (a) to about 900 in the bottom right of figure (c), with a median of 97.

distribution $p_\theta(\mathbf{z} \,|\, \mathbf{x})$ is generally intractable, so it is approximated using an inference model, or *encoder*, $q_\phi(\mathbf{z} \,|\, \mathbf{x})$. Analogously, the model $p_\theta(\mathbf{x} \,|\, \mathbf{z})$ is often referred to as the *decoder*. The VAE is trained by maximizing the lower bound on the evidence (ELBO), see (5), since

$$\log p_\theta(\mathbf{x}) \geq \mathbb{E}_{q_\phi(\mathbf{z}\,|\,\mathbf{x})} \left[ \log p_\theta(\mathbf{x}, \mathbf{z}) - \log q_\phi(\mathbf{z} \,|\, \mathbf{x}) \right] \tag{4}$$

$$= -D_{\text{KL}}(q_\phi(\mathbf{z} \,|\, \mathbf{x}) \,\|\, p(\mathbf{z})) + \mathbb{E}_{q_\phi(\mathbf{z}\,|\,\mathbf{x})} \left[ \log p_\theta(\mathbf{x} \,|\, \mathbf{z}) \right], \tag{5}$$

with $D_{\text{KL}}(\cdot \,\|\, \cdot)$ the Kullback-Leibler (KL) divergence [46]. By choosing a simple distribution for the encoder $q_\phi(\mathbf{z} \,|\, \mathbf{x})$, such as a multivariate Gaussian, the KL divergence has a closed-form expression, resulting in an efficient training algorithm.

We use importance sampling on the decoder [47] to approximate $\log p_\theta(\mathbf{x}_i)$ for the computation of the memorization score, and focus on the MNIST [48], CIFAR-10 [49], and CelebA [50] data sets. We use a fully connected encoder and decoder for MNIST and employ convolutional architectures for CIFAR-10 and CelebA. For the optimization we use Adam [51] and we implement all models in PyTorch [52]. The memorization score is estimated using $L = 10$ repetitions and $K = 10$ folds. Additional details of the experimental setup and model architectures can be found in Supplement B.

### 4.2 Results

We first explore memorization qualitatively. Figure 2 shows examples of observations with low, median, and high memorization scores in the VAE model trained on CIFAR-10. While some of the highly-memorized observations may stand out as odd to a human observer, others appear not unlike those that receive a low memorization score. This shows that the kind of observations that are highly memorized in a particular model may be counterintuitive, and are not necessarily visually anomalous.

If highly memorized observations are always given a low probability when they are included in the training data, then it would be straightforward to dismiss them as outliers that the model recognizes as such. However, we find that this is not universally the case for highly memorized observations, and a sizable proportion of them are likely *only* when they are included in the training data. If we consider observations with the 5% highest memorization scores to be "highly memorized", then we can check how many of these observations are considered likely by the model when they are included in the training data. Figure 3a shows the number of highly memorized and "regular" observations for bins of the log probability under the VAE model for CelebA, as well as example observations from both groups for different bins. Moreover, Figure 3b shows the proportion of highly memorized observations in each of the bins of the log probability under the model. While the latter figure shows that observations with low probability are *more likely* to be memorized, the former shows that a considerable proportion of highly memorized observations are *as likely as regular observations* when they are included in the training set. Indeed, more than half the highly memorized observations fall within the central 90% of log probability values (i.e., with $\log P_{\mathcal{A}}(\mathbf{x} \,|\, \mathcal{D}) \in [-14500, -12000]$).

The memorization score can be a useful diagnostic tool to evaluate the effect of different hyperparameter settings and model architectures. For example, in Figure 4c we illustrate the distribution of the memorization score for a VAE trained on MNIST with two different learning rates, and we show the train and test set losses during training in Figure 4a. With a learning rate of $\eta = 10^{-3}$ (blue curves), a clear generalization gap can be seen in the loss curves, indicating the start of overtraining (note the test loss has not yet started to *increase*). This generalization gap disappears when training with the smaller learning rate of $\eta = 10^{-4}$ (yellow curves). The absence of a generalization gap is

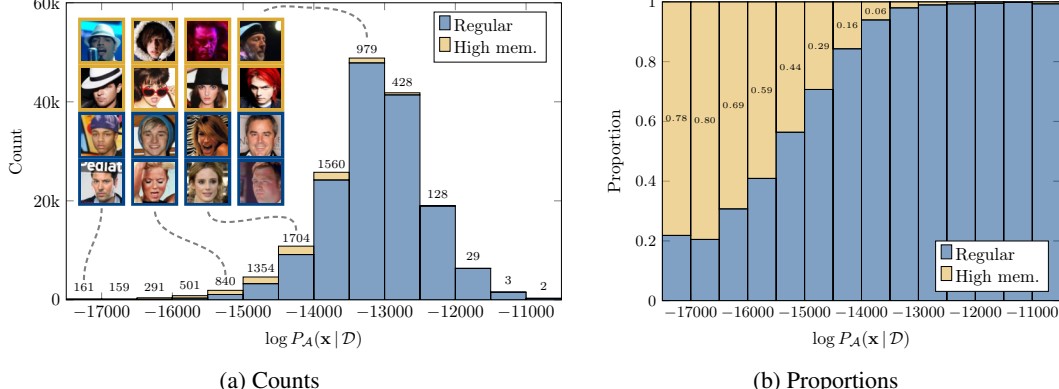

| (a) Counts | (b) Proportions |

Figure 3: In (a) we show a histogram of the number of highly memorized (yellow) and regular (blue) observations for bins of the log probability under a VAE model trained on the CelebA data set (where $n = 162,770$). The numbers above the bars correspond to the number of highly memorized observations in each bin. Randomly selected training observations from several bins are shown as well, with dashed lines illustrating the bin where the images in a particular column can be found. Images with a yellow frame are highly memorized whereas those with a blue frame have low memorization scores. Figure (b) shows the *proportion* of highly memorized and regular observations for each bin.

sometimes used as evidence for the absence of overfitting and memorization [53], but the distribution of the memorization scores for $\eta = 10^{-4}$ in Figure 4c shows that this is insufficient. While the memorization scores are *reduced* by lowering the learning rate, relatively high memorization values can still occur. In fact, the largest memorization score for $\eta = 10^{-4}$ is about 80, and represents a shift from a far outlier when the observation is absent from the training data to a central inlier when it is present.[3]

## 4.3 Memorization during training

To continue on the relation between memorization and overtraining, we look at how the memorization score evolves during training. In Figure 4b we show the 0.95 and 0.999 quantiles of the memorization score for the VAE trained on MNIST using two different learning rates. The quantiles are chosen such that they show the memorization score for the highly memorized observations. For both learning rates we see that the memorization score quantiles increase during training, as can be expected. However we also see that for the larger learning rate of $\eta = 10^{-3}$ the memorization score quantiles already take on large values before the generalization gap in Figure 4a appears. This is additional evidence that determining memorization by the generalization gap is insufficient, and implies that early stopping would not fully alleviate memorization. Moreover, we see that the rate of increase for the peak memorization quantiles slows down with more training, which suggests that the memorization score stabilizes and does not keep increasing with the training epochs. This is reminiscent of [20], who demonstrated that their metric for memorization in language models peaks when the test loss starts to increase. The difference is that here memorization appears to stabilize even before this happens.

## 4.4 Nearest Neighbors

As discussed in the introduction, nearest neighbor illustrations are commonly used to argue that no memorization is present in the model. Moreover, hypothesis tests and evaluation metrics have been proposed that measure memorization using distances between observations and model samples [26, 27]. Because of the prevalence of nearest neighbor tests for memorization, we next demonstrate the relationship between our proposed memorization score and a nearest neighbor metric.

As an example of a nearest neighbor test, we look at the relative distance of observations from the training set to generated samples and observations from the validation set. Let $\mathcal{S} \subseteq \mathcal{X}$ be a

---

[3]For this particular observation, $\log P_{\mathcal{A}}(\mathbf{x}_i \mid \mathcal{D}_{[n] \setminus \{i\}}) \approx -178$ when the observation is excluded from the training data, and $\log P_{\mathcal{A}}(\mathbf{x}_i \mid \mathcal{D}) \approx -97$ when it is included, and the latter value is approximately equal to the average log probability of the other observations with the same digit.

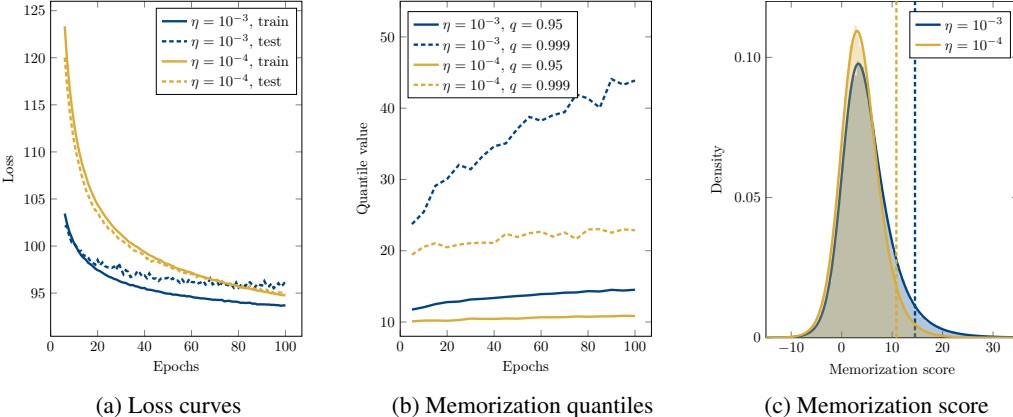

|                   |                    |                    |
|-------------------|--------------------|--------------------|
| (a) Loss curves   | (b) Memorization quantiles | (c) Memorization score |

Figure 4: Loss curves, memorization score quantiles, and memorization score distributions for a VAE trained on MNIST with different learning rates, $\eta = 10^{-3}$ (blue) and $\eta = 10^{-4}$ (yellow). In (c) the vertical lines mark the 95th percentile of the memorization scores, and the axes are cropped for clarity (the maximum memorization score for $\eta = 10^{-3}$ is about 210 and for $\eta = 10^{-4}$ it is about 80).

set of samples from the model and let $\mathcal{V} \subseteq \mathcal{X}$ be the validation set, with $|\mathcal{S}| = |\mathcal{V}|$. Denote by $d : \mathcal{X} \times \mathcal{X} \to \mathbb{R}$ a distance metric, which we choose here to be the Euclidean distance between images in pixel space after downsampling by a factor of 2. For all $\mathbf{x}_i$ in the training set $\mathcal{D}$, we then compute the ratio between the closest distance to a member of the validation set and a member of the sample set,

$$\rho_i = \frac{\min_{\mathbf{x} \in \mathcal{V}} d(\mathbf{x}_i, \mathbf{x})}{\min_{\mathbf{x} \in \mathcal{S}} d(\mathbf{x}_i, \mathbf{x})}. \tag{6}$$

If $\rho_i > 1$, then the nearest neighbor of $\mathbf{x}_i$ in the sample set is closer than the nearest neighbor in the validation set, and vice versa. Thus $\rho_i > 1$ suggests memorization is occurring, but as it depends on sampling it is expected to be very noise at an individual data point. Investigating if the *average* ratio for a set of observations differs significantly from 1 is an example of using hypothesis testing approaches to measure memorization.

Figure 5 illustrates the relationship between $\rho_i$ and the memorization score $M_i^{\text{K-fold}}$. We see that in general there is no strong correlation between the two score functions, which can be explained by the fact that they measure different quantities. While the memorization score directly measures how much the model relies on the presence of $\mathbf{x}_i$ for the local probability density, nearest neighbor methods test how "close" samples from the model are to the training or validation data. They thus require a meaningful distance metric (which is non-trivial for high-dimensional data) and are subject to variability in the sample and validation sets. We therefore argue that while nearest neighbor examples and hypothesis tests can be informative and may detect global memorization, to understand memorization at an instance level the proposed memorization score is to be preferred.

## 5 Mitigation Strategies

We describe two strategies that can be used to mitigate memorization in probabilistic generative models. First, the memorization score can be directly related to the concept of Differential Privacy (DP) [54, 55]. Note that the memorization score in (2) can be rewritten as

$$P_{\mathcal{A}}(\mathbf{x}_i \,|\, \mathcal{D}) = \exp(M_i^{\text{LOO}}) P_{\mathcal{A}}(\mathbf{x}_i \,|\, \mathcal{D}_{[n] \setminus \{i\}}), \tag{7}$$

and recall that a randomized algorithm $\mathcal{A}$ is $\varepsilon$-differentially private if for *all* data sets $\mathcal{D}_1, \mathcal{D}_2$ that differ in only one element the following inequality holds

$$P_{\mathcal{A}}(\mathcal{W} \,|\, \mathcal{D}_1) \le \exp(\varepsilon) P_{\mathcal{A}}(\mathcal{W} \,|\, \mathcal{D}_2), \quad \forall \mathcal{W} \subseteq \mathcal{X}. \tag{8}$$

Since this must hold for all subsets $\mathcal{W}$ of $\mathcal{X}$, it must also hold for the case where $\mathcal{W} = \{\mathbf{x}_i\}$. Moreover, when $\mathbf{x}_i$ is removed from $\mathcal{D}$ it can be expected that the largest change in density occurs at $\mathbf{x}_i$. It then follows that the memorization score can be bounded by employing $\varepsilon$-DP estimation techniques when

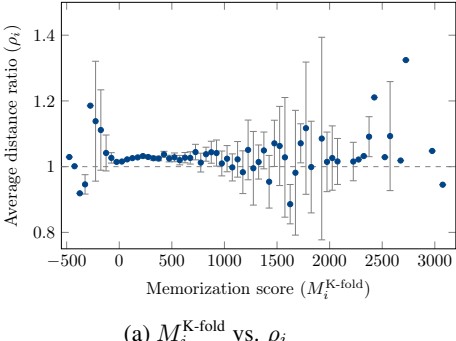
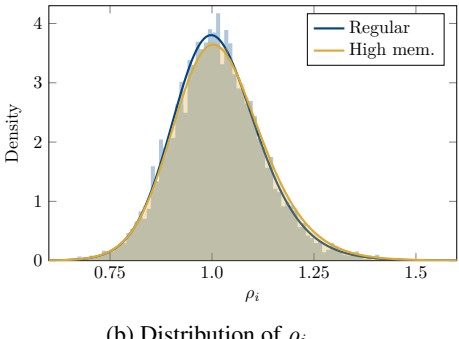

(a) $M_i^{\text{K-fold}}$ vs. $\rho_i$            (b) Distribution of $\rho_i$

Figure 5: Nearest neighbor distance ratio in (6) compared to the memorization score for a VAE trained on CelebA. Due to the large number of observations, we present the average of $\rho_i$ for bins of the memorization score of width 50, and show error bars representing the confidence interval of the standard error of the mean of distance ratio measurements in each bin. The horizontal axis in figure (a) trims off one observation at $M_i^{\text{K-fold}} \approx 6500$ for clarity. Figure (b) shows the distribution of the distance ratio for observations with a high memorization score (top 5%) and the regular ones.

training the generative model, as this will guarantee that $M_i^{\text{LOO}} \leq \varepsilon, \forall i$. The converse is however not true: observing a maximum memorization score of $M_i^{\text{LOO}} = \varepsilon$ for a particular model does not imply that the model is also $\varepsilon$-DP. This connection of the memorization score to differential privacy offers additional support for the proposed formulation of the memorization score.

An alternative approach to limit memorization is to explicitly incorporate an outlier component in the model that would allow it to ignore atypical observations when learning the probability density. This technique has been previously used to handle outliers in factorial switching models [56] and to perform outlier detection in VAEs for tabular data [57]. The intuition is that by including a model component with broad support but low probability (such as a Gaussian with high variance), the log probability for atypical observations will be small whether they are included in the training data or not, resulting in a low memorization score. Other approaches such as using robust divergence measures instead of the KL-divergence in VAEs [58] may also be able to alleviate memorization.

## 6 Discussion

We have introduced a principled formulation of a memorization score for probabilistic generative models. The memorization score directly measures the impact of removing an observation on the model, and thereby allows us to quantify the degree to which the model has memorized it. We explored how the memorization score evolves during training and how it relates to typical nearest neighbor tests, and we have shown that highly memorized observations are not necessarily unlikely under the model. The proposed memorization score can be used to determine regions of the input space that require additional data collection, to understand the degree of memorization that an algorithm exhibits, or to identify training observations that must be pruned to avoid memorization by the model.

A question that requires further study is what constitutes a "high" memorization score on a particular data set. One of the main difficulties with this is that density estimates returned by a model, and thus probability differences, are not necessarily comparable between data sets [34]. We expect that future work will focus on this important question, and suggest that inspiration may be taken from work on choosing $\varepsilon$ in differential privacy [59, 60]. Furthermore, exploring the relationship between memorization and the double descent phenomenon [61, 62] could be worthy of investigation. Improving the efficiency of the estimator is also considered an important topic for future research.

If we want diversity in the samples created by generative models, then the model will have to learn to generalize to regions of the data manifold that are not well represented in the input data. Whether this is achieved by extrapolating from other regions of the space or fails due to memorization is an important question. Our work thus contributes to the ongoing effort to understand the balance between memorization and generalization in deep generative neural networks.

## Acknowledgments and Disclosure of Funding

The authors would like to thank the anonymous reviewers for their helpful comments, and members of the AIDA project for helpful discussions. This work was supported in part by The Alan Turing Institute under EPSRC grant EP/N510129/1.

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
