# On Memorization in Probabilistic Deep Generative Models
## Supplementary Material

## A  Memorized observations in recently proposed generative models

While experimenting with the proposed memorization score on CIFAR-10 [47], we noticed that the images of automobiles shown in Figure 6 are present in the training set multiple times (with slight variation). We subsequently spotted these images in the illustrations of generated samples in [7] (Figure 13, example (a) can be seen twice) and [61] (Figure 11 and Figure 13, truck class). These works are recently proposed probabilistic generative models that achieve impressive performance on sample quality metrics such as the inception score (IS) [35] and the Fréchet inception distance (FID) [36], and also achieve high log likelihoods. However, the fact that we were able to serendipitously spot images from the training set in the generated samples might suggest that some unintended memorization occurs in these models. We do not know if there are other images in the presented samples that are present in the training data.

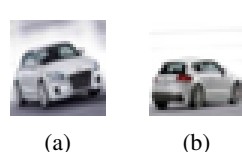

(a)                (b)

Figure 6: Examples of images from the CIFAR-10 training set that were spotted in illustrations of samples from the model in recent work on generative models.

Of course, spotting near duplicates of training observations is only possible because these models yield realistic samples. As we argue in the main text and as has been shown by previous works [32, 37], quality metrics such as IS and FID do not detect memorization.

We emphasize that this evidence is presented mainly to support the notion that (unintended) memorization can occur in probabilistic deep generative models, and to provide additional motivation for understanding and quantifying when and how memorization arises, which is the focus of our work.

## B  Experimental Details

This section describes additional details of the data sets, model architectures, and experimental setup.

### B.1  Datasets

We use the MNIST [46], CIFAR-10 [47], and CelebA [48] data sets, which are widely used and are freely available for research purposes (although to the best of our knowledge explicit licenses are not available). For MNIST we binarize the images dynamically during training by considering each grayscale pixel value as the parameter of an independent Bernoulli variable, as is common [11, 62]. Images in all data sets are resized to $32 \times 32$ pixels for efficiency and ease of implementation. CIFAR-10 contains color images from 10 different categories and does not require further preprocessing. CelebA contains potentially identifiable images of faces of celebrities sourced from publicly available images on the Internet. We used the predefined cropping function of [11] to center the face region. For CIFAR-10 and CelebA we used random horizontal flips during training as data augmentation. All data sets have predefined train and test sets, and CelebA additionally has a validation set. We mainly used the training sets in the experiments, with the exception of the experiments for Figure 4a, which uses the MNIST test set, and the experiments in Section 4.4, which use the CelebA validation set.

### B.2  Model Architectures

Let $L$ denote the size of the latent space and recall that $\mathbf{x} \in \mathcal{X} \subseteq \mathbb{R}^D$. For all experiments we used a Gaussian encoder with a learned diagonal covariance matrix, $q_\phi(\mathbf{z} \,|\, \mathbf{x}) = \mathcal{N}(\mathbf{z}; \boldsymbol{\mu}_\phi(\mathbf{x}), \mathrm{diag}(\boldsymbol{\sigma}^2_\phi(\mathbf{x})))$ and a standard multivariate Gaussian prior on the latent variables, $p(\mathbf{z}) = \mathcal{N}(\mathbf{z}; 0, \mathbf{I}_L)$. As mentioned above we used a dynamically binarized version of the MNIST data set, and therefore used a Bernoulli likelihood for the decoder of the VAE. Both the encoder and decoder used fully connected layers with the RELU activation on the intermediate layers [63] and a sigmoid activation on the output of the decoder that represents the parameter of the Bernoulli distribution. For MNIST we used $L = 16$. Full details of the model architecture are given in Table 1.

For CIFAR-10 and CelebA we used a Gaussian likelihood for the decoder, employed uniform dequantization on the pixel values [64], and trained the models in logit space following [65]. For both data sets

Table 1: Model architectures used for the experiments. We used fully connected (FC) layers with the RELU activation for MNIST, with the SIGMOID activation on the decoder. For CIFAR-10 and CelebA we used convolutional layers for the encoder (CONV2D with kernel size 4, stride 2, and padding 1), followed by batch normalization (BN), and the Leaky ReLU activation (LRELU, using slope 0.2). For these data sets the decoder consists of transposed convolution layers (CONVT2D, with kernel size 4, stride 2, and padding 1 except for the layer marked with an asterisk (*), which uses kernel size 2, stride 1, and padding 0 to get the correct output size), followed by batch norm and the ReLU activation. We use the abbreviations $\text{ENCBLOCK}(C_1, C_2) = \text{CONV2D}(C_1, C_2) \to \text{BN} \to \text{LRELU}$ and $\text{DECBLOCK}(C_1, C_2) = \text{CONVT2D}(C_1, C_2) \to \text{BN} \to \text{RELU}$.

| Data set | Encoder network | Decoder network | Likelihood $(p_\theta(\mathbf{x} \mid \mathbf{z}))$ |
|---|---|---|---|
| MNIST | $\text{FC}(1024, 512) \to \text{RELU}$ $\to \text{FC}(512, 256) \to \text{RELU}$ $\to \text{FC}(256, L), \text{FC}(256, L)$ | $\text{FC}(L, 256) \to \text{RELU}$ $\to \text{FC}(256, 512) \to \text{RELU}$ $\to \text{FC}(512, 1024)$ $\to \text{SIGMOID}$ | $\mathcal{B}(x_{ij}; \pi_{ij}(\mathbf{z}))$ |
| CIFAR-10 | $\text{CONV2D}(C, F) \to \text{LRELU}$ $\to \text{ENCBLOCK}(F, 2F)$ $\to \text{ENCBLOCK}(2F, 4F)$ $\to \text{ENCBLOCK}(4F, 8F)$ $\to \text{FLATTEN}$ $\to \text{FC}(32F, L), \text{FC}(32F, L)$ | $\text{DECBLOCK}^*(L, 8F)$ $\to \text{DECBLOCK}(8F, 4F)$ $\to \text{DECBLOCK}(4F, 2F)$ $\to \text{DECBLOCK}(2F, F)$ $\to \text{CONVT2D}(F, 2C)$ | $\mathcal{N}(\mathbf{x}; \boldsymbol{\mu}_\theta(\mathbf{z}), \text{diag}(\boldsymbol{\sigma}_\theta(\mathbf{z})))$ |
| CelebA | Same as for CIFAR-10 | Same as for CIFAR-10, except final layer uses $\text{CONVT2D}(F, C)$ | $\mathcal{N}(\mathbf{x}; \boldsymbol{\mu}_\theta(\mathbf{z}), \gamma_\theta \mathbf{I}_D)$ |

we used an architecture similar to DCGAN [66], consisting of four convolutional layers in the encoder, each followed by batch normalization [67] and leaky RELU activation [68], and five transposed convolution layers in the decoder followed by batch normalization and RELU, see Table 1. For CIFAR-10 the Gaussian likelihood on the decoder was parameterized as $p_\theta(\mathbf{x} \mid \mathbf{z}) = \mathcal{N}(\mathbf{x}; \boldsymbol{\mu}_\theta(\mathbf{z}), \text{diag}(\boldsymbol{\sigma}_\theta(\mathbf{z})))$ and for CelebA we used the simpler formulation $p_\theta(\mathbf{x} \mid \mathbf{z}) = \mathcal{N}(\mathbf{x}; \boldsymbol{\mu}_\theta(\mathbf{z}), \gamma_\theta \mathbf{I}_D)$ with a learned parameter $\gamma_\theta$, as the more general decoder was unnecessary. For CIFAR-10 and CelebA the number of input channels is $C = 3$ and we used $L = 64$ and $L = 32$, respectively. For the convolutional networks the feature map multiplier was set to $F = 32$ (see Table 1).

## B.3 Training details

We used Adam [49] to optimize the parameters of the model with learning rate $\eta = 10^{-3}$ for the main experiments and $\eta = 10^{-4}$ for the experiments on MNIST in Section 4.2. We used a batch size of 64 and left the remaining parameters for Adam at their default values in PyTorch [50]. For both MNIST and CIFAR-10 we trained for 100 epochs, and used 50 epochs for CelebA. These settings were chosen by taking into consideration the available computational resources and aimed to avoid overtraining. The parameter settings were determined through some preliminary experimentation and were not extensively optimized. Experiments were conducted on a desktop machine running Arch Linux, using an NVIDIA GeForce GTX 1660 SUPER GPU, 32GB of RAM, and an AMD Ryzen 5 3600 processor. Total wall-clock time was about 200 hours for the main results, excluding preliminary experimentation. Electricity needed for the experiments came from carbon-free sources.

As mentioned in the main text, importance sampling was used to approximate $p(\mathbf{x})$, such that

$$p(\mathbf{x}) \approx \frac{1}{N} \sum_{l=1}^{N} \frac{p_\theta(\mathbf{x} \mid \mathbf{z}_l) p(\mathbf{z}_l)}{q_\phi(\mathbf{z}_l \mid \mathbf{x})}, \qquad \mathbf{z}_l \sim q_\phi(\mathbf{z} \mid \mathbf{x}). \tag{9}$$

This was computed in log space for numerical accuracy. For MNIST we used $N = 256$ and for CIFAR-10 and CelebA we used $N = 128$ samples.

# C Additional Results

Below we show additional results that confirm the findings presented in the main text for different data sets.

## C.1 Qualitative Illustrations

In Figures 7, 8, and 9 we illustrate observations with low, median, and high memorization scores for a VAE trained on MNIST using $\eta = 10^{-3}$, MNIST using $\eta = 10^{-4}$, and CelebA, respectively. As can be seen from the figures and as discussed in the main text in Section 4.2, while some of the highly memorized observations have visual anomalies, others are not unlike those that receive low memorization scores. For instance, for the VAE trained on MNIST with learning rate $\eta = 10^{-4}$, we see that images from both the low and high memorization groups have active pixels that are not part of the digit (compare, for instance, the images of 9s on the middle of the bottom rows of Figure 8a and Figure 8c).

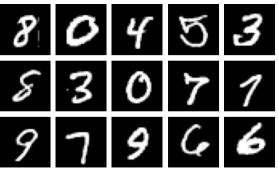 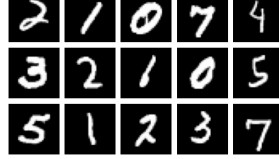 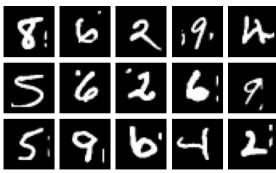

(a) Low memorization      (b) Median memorization      (c) High memorization

Figure 7: Observations with low, median, and high memorization scores in the MNIST data set, for a VAE trained using learning rate $\eta = 10^{-3}$. Memorization scores range from about $-18$ in the top left of figure (a) to about 200 in the bottom right of figure (c), with a median of 4.4.

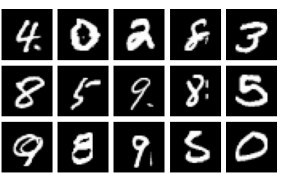                      

(a) Low memorization      (b) Median memorization      (c) High memorization

Figure 8: Observations with low, median, and high memorization scores in the MNIST data set, for a VAE trained using learning rate $\eta = 10^{-4}$. Memorization scores range from about $-13$ in the top left of figure (a) to about 80 in the bottom right of figure (c), with a median of 3.5.

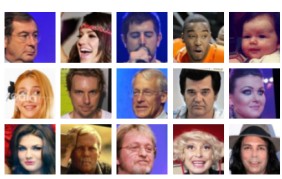                    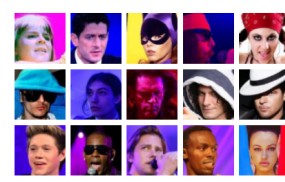

(a) Low memorization      (b) Median memorization      (c) High memorization

Figure 9: Observations with low, median, and high memorization scores in the CelebA data set when the density is learned using a convolutional VAE. Memorization scores range from about $-450$ in the top left of figure (a) to about 6500 in the bottom right of figure (c), with a median of about 60.

## C.2 Outliers vs. Memorization

Figures 10 and 11 replicate the experiments shown in Figure 3 in the main text for the VAE trained on the MNIST data set using two different learning rates. We again see that relatively high memorization is not exclusive to observations that receive a low probability under the model. Note that for this particular data set the density estimated by the VAE is slightly multimodal, with the peak in density for higher values of $\log P_\mathcal{A}(\mathbf{x} \mid \mathcal{D})$ corresponding to observations for digit 1.

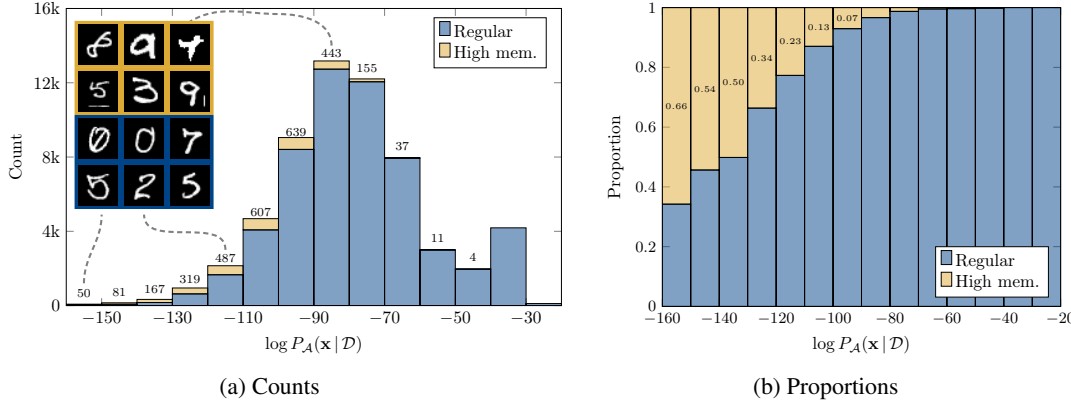

(a) Counts

(b) Proportions

Figure 10: In (a) we show a histogram of the number of highly memorized (yellow) and regular (blue) observations for bins of the log probability under a VAE model trained on the MNIST data set using learning rate $\eta = 10^{-3}$. The numbers above the bars correspond to the number of highly memorized observations in each bin (for MNIST, $n = 60,000$). Randomly selected training observations from several bins are shown, with dashed lines illustrating the bin where the images in a particular column can be found. Images with a yellow frame are highly memorized whereas those with a blue frame have low memorization scores. Figure (b) shows the *proportion* of highly memorized and regular observations for each bin.

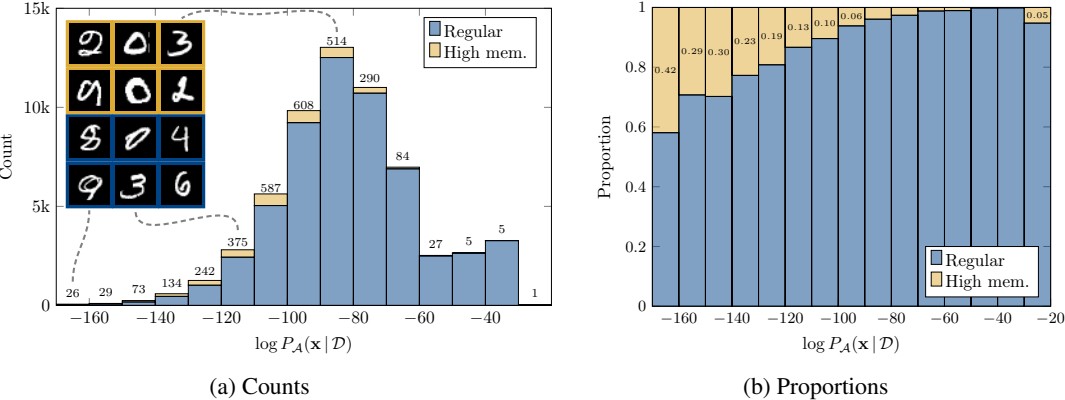

(a) Counts

(b) Proportions

Figure 11: Similar to Figure 10, but for the VAE trained on MNIST with learning rate $\eta = 10^{-4}$.

## C.3 Nearest Neighbors

The nearest neighbor experiments demonstrated in Section 4.4 are repeated below in Figures 12 and 13 for the VAEs trained on the MNIST data set using learning rates of $10^{-3}$ and $10^{-4}$. For these models and data set we again do not see a clear relation between the nearest neighbor distance ratio $\rho_i$ and the proposed memorization score $M_i^{\text{K-fold}}$, as discussed in the main text.

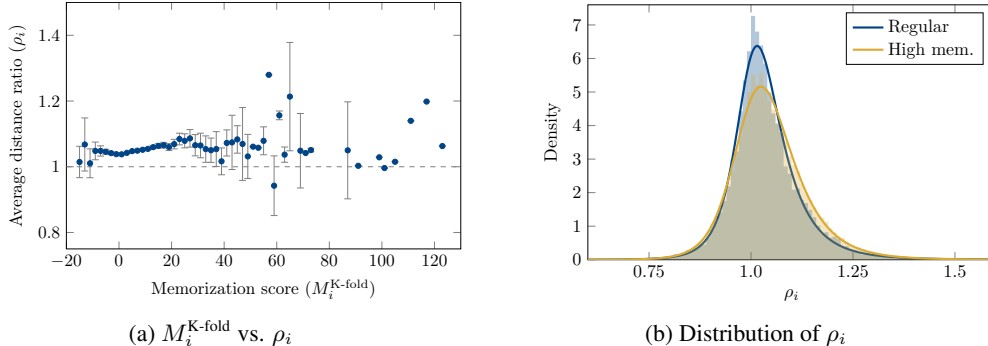

(a) $M_i^{\text{K-fold}}$ vs. $\rho_i$            (b) Distribution of $\rho_i$

Figure 12: Illustration of the nearest neighbor distance ratio in (6) compared to the memorization score for a VAE trained on MNIST using a learning rate of $\eta = 10^{-3}$. We present the average of $\rho_i$ for bins of the memorization score of width 2, and show error bars representing the confidence interval of the standard error of the mean of distance ratio measurements in each bin. The horizontal axis in figure (a) trims off one observation at $M_i^{\text{K-fold}} \approx 210$ for clarity. Figure (b) shows the distribution of the distance ratio for observations with a high memorization score (top 5%) and the regular ones.

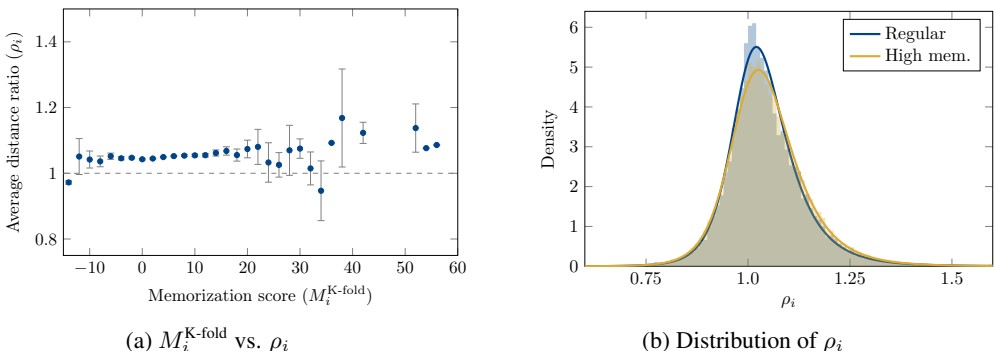

(a) $M_i^{\text{K-fold}}$ vs. $\rho_i$            (b) Distribution of $\rho_i$

Figure 13: Similar to Figure 12 but for the VAE trained on MNIST using a learning rate of $\eta = 10^{-4}$. The horizontal axis in figure (a) trims off one observation at $M_i^{\text{K-fold}} \approx 80$ for clarity.