# OpenReview forum: "On Memorization in Probabilistic Deep Generative Models"
_NeurIPS.cc/2021/Conference — NeurIPS 2021 Poster_

### Official Review · Reviewer_x5oB · 2021-07-05

**Rating:** 6
**Confidence:** 4

**Summary:**

The authors propose a novel metric to measure generalization in generative models. This metric is defined to be the ratio between  the log probability of a data sample when it was included in the training set and its logP when it was not included in the training set. The authors show that this metric is reasonable and show some interesting properties of the learning dynamics of generative models that become apparent when tracking the memorization score.

**Limitations And Societal Impact:**

The authors discuss the limitation of their work, but have not addressed any potential negative societal impact for their work. Adding a paragraph to that effect would be useful.

**Main Review:**

The writing is clear and easy to follow. The idea is simple and, while it is adapted from another paper on supervised learning, is being applied to a novel enough domain to have relevance for further research. Furthermore, the authors illustrate interesting points about the learning dynamics of generative models such as the fact that memorization happens early on in training and plateaus, and that memorization does not seem to be correlated with the train test generalization gap. These findings are consistent with the work around double descent, where it was proposed that the double descent phenomenon occurs as networks "memorize" training data early on in training, and then improve performance as they move to an interpolation regime. Indeed, one would expect  larger models to "memorize" more and faster, and it would be interesting to see such an experiment in this paper.

A few negative aspects:
- I am not sure what point the authors are trying to make with the comparison with nearest neighbors. They argue (correctly) that their score and NN are measuring two different aspects of memorization, but I don't see what value this long discussion adds to the paper. This connection could probably be explained more briefly and the paper expended with more experiments :)
- Related to the above, I was disappointed to see an interesting connection presented in section 5 between Differential Privacy and memorization, and how it could be used to address memorization in generative models, but no experimental results of any kind to show this point. This confuses the reader as to why this is a separate section.

**Time Spent Reviewing:**

2

---

> ### Author Response · Authors · 2021-08-09
> **Response to reviewer x5oB**
>
> We thank the reviewer for the kind words and are encouraged to see they found our work interesting, novel, and relevant, and we are pleased to note that the writing was considered clear and easy to follow.
>
> > These findings are consistent with the work around double descent
>
> We are intrigued by the connection made to the double-descent phenomenon. While space limitations likely prohibit us from exploring this in detail in the present paper, we will mention this connection in the revised manuscript and recommend it as a direction for future research. Although model-wise double descent is certainly worth exploring, we do note that Fig. 10 of Nakkiran et al (2019) suggests that epoch-wise double descent does not occur in the absence of label noise (which may be more comparable to the unsupervised setting).
>
> > * I am not sure what point the authors are trying to make with the comparison with nearest neighbors.
>
> The two main reasons for making the connection to nearest neighbor tests are that there are several existing works that propose using nearest neighbor tests to measure memorization, and nearest neighbor figures are often presented in work on generative models as an argument against the presence of memorization. We therefore consider it important to explore the (lack of) correspondence between our proposed memorization score and such nearest neighbor tests.
>
> > * I was disappointed to see an interesting connection presented in section 5 [...] but no experimental results of any kind to show this point.
>
> We focused our experimental efforts on understanding the memorization phenomenon in probabilistic deep generative models and how it relates to the generalization gap and nearest neighbor tests. Because of the clear theoretical connection between our memorization score and differential privacy, and because differential privacy has been shown to limit other forms of memorization such as membership inference (e.g. Hayes et al. 2018), we chose not to pursue such experiments in this work. We do however consider them important topics for future research and will emphasize this in the paper.

---

> > ### Comment · Reviewer_x5oB · 2021-08-31
> > **Author response**
> >
> > I thank the authors for their answers to my queries.
> >
> > I am still convinced that the paper would be made stronger by adding a few more experimental results that strengthen the utility of the idea explained here. First, the connection between differential privacy and memorization as originally requested. Second, experiments testing how memorization is impacted by model scaling and double descent phenomena could help to address the concerns of reviewer 3dmL around the distinction between overfitting and memorization, by highlighting that we might want to distinguish between differently "overfit" models, and we can use the memorization score to do that.

---

> > > ### Author Response · Authors · 2021-08-31
> > > **Response to reviewer x5oB**
> > >
> > > Thank you for your comments.  Please see our latest comment to reviewer 3dmL regarding the connection between memorization and overfitting.

---

### Official Review · Reviewer_cz89 · 2021-07-10

**Rating:** 6
**Confidence:** 4

**Summary:**

This paper investigates how probabilistic deep generative models memorize training samples. They demonstrate that highly memorized observations are not necessarily outliers, that memorization occurs early in the training process, and investigate existing nearest neighbor tests for memorization on MNIST, CIFAR-10, and CelebA.

**Limitations And Societal Impact:**

- The authors discuss the computational barriers to implementing their proposed memorization score, and propose alternative approximations that are much more tractable in Section 3. [Do they discuss the impact of t and K beyond Section 4.1/Appendix B?] They also provide guidelines for future investigations into comparing memorization scores across datasets.
- Limiting memorization in deep generative models is important, as it is often undesirable for synthetic samples to directly mimic training examples (for privacy concerns, for example). But the authors should also provide a discussion on potential negative societal consequences of their work.


**Main Review:**

Originality:
- This paper extends a recently proposed method for assessing memorization in neural networks (Feldman et al. 2019, Feldman et al. 2020) for generative models. The authors make it clear that the problem of memorization is different in a density estimation setting rather than a classification setting, and they adequately addressed related work in Section 2.

Quality:
- The claims in the paper were technically sound and the method made sense to me.
- However, I think what is missing is a comparison of how existing metrics for generative models (e.g. inception score, Frechet inception distance, Kernel inception distance, etc.), model architectures, model families, etc. affect memorization. Although there was a brief discussion about it in l.112-113, the paper would be much stronger if the authors could explore how we can better leverage this memorization score to draw conclusions about a trained model (or to compare sets of trained models). I’ve talked more about this point in the “Significance” section.
- Additionally, Section 5 (“Mitigation Strategies”) doesn’t actually explore how to mitigate memorization in deep generative models, and suggests a few strategies for doing so which have yet to be explored.

Clarity:
- The paper is well-written and easy to read, which I appreciated. The authors provide adequate details for their experiments both in the main text and supplementary material and have included source code to replicate their results.

Significance:
- Memorization in deep generative models has not been explored very much, and is an important area of research. The authors’ results are interesting because their memorization score allows for quantifying whether a single observation has been memorized in the data. I liked their comparison to existing approaches (e.g. nearest neighbors tests), and how they demonstrated that their method was capturing a different kind of signal.
- I would have appreciated a further discussion on the implications of the results. One question that I am left with is how exactly I should use this memorization score in practice. Is it more of an exploratory tool to get a sense of which examples are being memorized by my generative model? Do certain kinds of generative models (e.g. VAEs with better samples, for example according to FID score, or GANs vs. VAEs) lead to higher memorization? Given the introduction of the memorization score and the suggestions in Section 5, would the authors demonstrate how to actually reduce memorization in these models (for example, to take a model that has high memorization and to lower its memorization score)? Answers to such questions would greatly strengthen the paper.

Comments/Questions:
- What are “explicit memory cells” in language models that can facilitate memorization in l.85? Is this referring to attention/LSTMs? Technically language models are a type of generative model.
- I’m curious as to why the authors didn’t investigate GANs in more detail, since they are more prone to mode collapse and the memorization problem relative to VAEs. Was the idea to focus on likelihood-based models?
- The phrase “using importance sampling on the decoder” is confusing (l. 194). I see that in the supplementary material the authors were referring to importance sampling of the latent z’s in the approximate posterior -- this should be changed in the main text.
- I’m not sure what is the distinction between “global memorization” (which can be detected by nearest neighbor tests) and “instance-level memorization” (as per the proposed memorization score) in l.268-269. Would the authors elaborate between the two?
- What is an “outlier component” in l. 283? Would it be a loss term that prevents the model from learning from outliers/discards them during training?
- As per Figure 2, would it be safe to say that the model memorizes highly “atypical” training set examples, such as the dog with the neon green background?

Minor typos:
- z \in \mathcal{Z} \in \mathbb{R}^L in l. 186
- Should Line 10 in Algorithm 1 be V_i - U_i as per Eq. 2?

----------------------------
UPDATE: Thank you to the authors for their response, it was helpful for clarifying some points that I was confused about. I've also read the other reviewers' responses. I think this work tries to tackle a sufficiently important, understudied problem and will be nice for the community to build on, so I've increased my score to a 6.

**Time Spent Reviewing:**

3

---

> ### Author Response · Authors · 2021-08-09
> **Response to reviewer cz89**
>
> We thank the reviewer for the detailed feedback on our manuscript and are glad to note that the reviewer found our work interesting, technically sound, and well-written.
>
> > * I would have appreciated a further discussion on the implications of the results.
>
> We believe the memorization score can be a useful tool to diagnose the learning dynamics of probabilistic generative models, and complements existing quality metrics such as the IS and FID (which, as we note, can't detect memorization). While our present work focused on establishing the memorization score and gaining an understanding of how memorization arises in practice (as well as how it relates to the generalization gap, nearest neighbor tests, etc.), we certainly agree that evaluating the memorization properties of different kinds of generative models is an important topic for future research.
>
> > * Given the introduction of the memorization score and the suggestions in Section 5, would the authors demonstrate how to actually reduce memorization in these models [...]?
>
> In this work we have focused our experimental efforts on understanding memorization and have not experimented with the proposed mitigation strategies, although we consider it an important topic for future work. This is in part because we have the clear theoretical connection of our memorization score to differential privacy, but also because differential privacy has already been shown to limit other forms of memorization, such as membership inference attacks (e.g. Hayes et al, 2018).
>
> > * What are "explicit memory cells"?
>
> We intended LSTM and GRU units with this comment, and will clarify this.
>
> > * I'm curious as to why the authors didn't investigate GANs in more detail
>
> We focused on explicit density models as this allowed us to define a principled memorization score, which is not possible for GANs. Moreover, while phenomena such as mode-collapse in GANs are broadly studied, memorization in probabilistic generative models is not. We will add a discussion on possibly adapting the memorization score to implicit-likelihood models such as GANs.
>
> > * I'm not sure what is the distinction between "global memorization" (which can be detected by nearest neighbor tests) and "instance-level memorization"
>
> The distinction here is that instance-level memorization can occur in the absence of global memorization. The latter is also typically a consequence of overtraining and thus affects the entire model, whereas we demonstrate that instance-level memorization can occur before overtraining happens.
>
> > * What is an "outlier component" in l. 283? Would it be a loss term that prevents the model from learning from outliers/discards them during training?
>
> Indeed the outlier component would be a feature of the model that assigns low probability to outliers and prevents them from affecting the model. We will expand on this in the text.
>
> > * As per Figure 2, would it be safe to say that the model memorizes highly "atypical" training set examples, such as the dog with the neon green background?
>
> This is certainly true, although we further demonstrate that memorization is not _exclusive_ to such atypical observations.
>
> > * Do they discuss the impact of t and K beyond Section 4.1/Appendix B?
>
> As mentioned in our response to Reviewer 4FjL, we will include an appendix to explore the bias, variance, and convergence of our estimator.
>
> > * The authors should also provide a discussion on potential negative societal consequences of their work.
>
> We will critically examine any negative societal consequences of our work and update our manuscript appropriately.

---

### Official Review · Reviewer_4FjL · 2021-07-16

**Rating:** 6
**Confidence:** 4

**Summary:**

This paper proposes a point-wise memorization score for generative models, which is given by the difference of log average (under some prior over hyperparameters) probabilities assigned to the point when the model is trained with and without the observation, respectively. The authors propose to estimate this score with a sample-based estimate reminiscent of cross validation, where the data is split into folds, and the model just has to be trained over the dataset minus each of the folds, rather than over the dataset minus every single datapoint. The authors then attempt to get insights into memorization in VAEs using their score.

**Ethical Concerns:**

None.

**Limitations And Societal Impact:**

Limitations, other than the variance one I mentioned above are properly discussed in the paper, as are potential negative societal consequences.

**Main Review:**

I think this paper addresses an interesting and potentially relevant yet understudied topic in generative modeling. Measuring memorization as changes in log likelihoods after excluding a point seems sensible, and I found the connection made to differential privacy particularly interesting. Additionally, the paper is well written and easy to follow.

My main concern with the proposed method is that the authors have no discussion whatsoever about the bias and variance of their proposed memorization score estimator, as their actual memorization score cannot be computed in closed form. While a theoretical analysis of these quantities would obviously be ideal, if not available, some sanity checks should at least be performed. For example, in lines 204-205, the authors claim "the kind of observations that are highly memorized in a particular model may be counterintuitive, and are not necessarily visually anomalous", which I think might also be explained by high variance of the estimate. Similarly, the quantiles in Figure 4b might be dominated by the variance of the estimates, rather than memorization itself. If the authors provide evidence of the robustness of their estimate during the rebuttal, I will increase my score.

In section 5, the authors discuss a relationship between their memorization score and differential privacy; and thus how methods from differential privacy could be used to reduce memorization. While there are some settings where one can clearly think why memorization would be bad (e.g. privacy, or extreme memorization of any single data point), it is not obvious to me that it is often so. In other words, is memorization often a problem we should care about in our current models? I believe the paper would benefit from a discussion about this.

======================================================================================================

EDIT 1 AFTER REBUTTAL

I have read the other reviews as well as the author's rebuttal, and have decided to increase my score. My concerns about the observed results being simply a consequence of high-variance estimates of the memorization score are diminished after the additional experiment done by the authors, even if I am still unsure that memorization can be problematic outside of settings where privacy matters.

======================================================================================================

**Time Spent Reviewing:**

4

---

> ### Author Response · Authors · 2021-08-09
> **Response to reviewer 4FjL**
>
> We thank the reviewer for their thoughtful review and are pleased to see they found our work interesting, sensible, and well-written.
>
> > My main concern with the proposed method is that the authors have no discussion whatsoever about the bias and variance of their proposed memorization score estimator
>
> We explored the bias of our estimator in a toy experiment using the Generative Topographic Mapping (GTM) probabilistic generative model (Bishop et al., 1998) and a dataset of 400 observations. In this setting we can compute the memorization score exactly and compare it to our proposed estimator. This experiment shows strong correspondence between the true and estimated memorization scores. Moreover, we see that the cross validation-based approach in eq. (3) closely matches the true value of the leave-one-out log probability. Please see the figures at this anonymized link: https://imgur.com/a/w0tDSDp We will include an appendix with a detailed analysis of the bias and variance of the estimator using this experiment.
>
> > In other words, is memorization often a problem we should care about in our current models?
>
> It is indeed the case that where data privacy matters, high memorization is a cause for concern. In other scenarios, we argue that high memorization can be informative for the modeller as it indicates that an observation may have an outsized effect on the local probability density (consider for instance the left-most blue observation in Fig. 1a). On inspection of such observations, a practitioner may want to collect more data for a particular region of the data manifold or decide to exclude certain images. Considering Figure 9c as an example, this could include images of individuals wearing hats or in front of atypical backgrounds. We will expand on this in the revised manuscript.

---

### Official Review · Reviewer_3dmL · 2021-07-16

**Rating:** 5
**Confidence:** 4

**Summary:**

This paper proposes a new measure to quantify memorization in generative models. The proposed measure is an adapted version of a memorization measure that was recently proposed for supervised learning models. The measure reflects how much the likelihood of a training sample under the trained model would decrease if that sample would be excluded from the training set. Using a VAE model as example, the authors show that training samples that are highly memorized according to their proposed measure are not necessarily samples with low likelihood (or outliers). It is further shown that the scores of the proposed memorization measure do not strongly correlate with memorization scores from a nearest neighbour test, and the authors discuss several strategies that could be used to “alleviate memorization”.

**Limitations And Societal Impact:**

I don’t have anything additional to add here, please see my above comment.

**Main Review:**

This paper is well-written; I read it with pleasure and great interest. The memorization measure proposed by the paper is clearly defined, intuitive, well-motivated and I believe it could be a useful diagnostic tool to inspect and better understand the behaviour of deep generative models. The proposed measure is not entirely original as it is adapted from a similar, recently proposed memorization measure for supervised learning, but the authors clearly state this and I do not think this diminishes the value or contribution of the current paper much.

Nevertheless, I’m afraid that I lean towards rejecting this paper, for the reasons listed below. I am however open to being challenged on these in the rebuttal phase, and to re-consider my score.

Firstly, a central claim of the paper is that memorization is fundamentally different from overfitting, but this distinction is not clear to me. It seems to me that both of these are on the same continuum. On this continuum one extreme is overfitting (corresponding to very high memorization scores) and the other extreme is an uninformative model that ignores the data (corresponding to a memorization score of 0). Let’s take Fig 1 as example. Panel (a) is an example of what the authors call memorization, and the model of this example thus presumably corresponds to relatively high memorization scores. Panel (b) is an example of what the authors call overfitting. Unless I’m mistaken, the model of this example will have higher memorization scores than the model from panel (a), suggesting that overfitting is (simply?) a more extreme version of memorization. I’m still happy to be convinced otherwise, but at the moment I do not see how these two concepts are fundamentally different.

Secondly, a central assumption of the paper seems to be that memorization in generative model is undesired and problematic, but this is again not clear to me. It is certainly not the case that a lower memorization score is always better, because a memorization score of 0 is clearly not desirable as it means that the model ignores the data. The authors seem to be mostly concerned about the question when their memorization score is problematically high; should we not also be concerned about the question when the memorization score is problematically low?

Thirdly, I think there is an important issue with the experiment described in the second paragraph of section 4.2 (related to Fig. 3). In this experiment it is shown that some of the most highly memorized examples are “as likely [under the regular model] as regular observations”. This is taken as evidence that highly memorized observations are not necessarily outliers. However, an important, unaddressed confound in this analysis is that the memorization score of an example is defined as its likelihood under the regular model minus its likelihood under the model fitted with that example left out. (That is, by definition, the higher the likelihood under the regular model, the higher the memorization score.)

Fourthly, an important limitation of the proposed memorization score is that its absolute value is hard to interpret. For example, it is not clear when a memorization score is high or problematic. It should be said that the authors clearly discuss this limitation in the paper (e.g., L165-175). However, in the final paragraph of section 4.2 (related to Fig. 4), the authors nevertheless claim that with the VAE trained with learning rate 10e-4 “high memorization can still occur”. The authors seem to justify this interpretation with a case study of a single data point with a high memorization score, but this is not convincing to me and I think it would be good if the authors could elaborate on this more.

Minor comment:
-	It might be useful to briefly discuss to what extent the proposed memorization measure could be applied to GANs.


**Time Spent Reviewing:**

7

---

> ### Author Response · Authors · 2021-08-09
> **Response to reviewer 3dmL**
>
>
> We thank the reviewer for the detailed comments and are glad to find that the reviewer read our manuscript with interest and found our work intuitive, well-motivated, and useful.
>
> > Firstly, a central claim of the paper is that memorization is fundamentally different from overfitting, but this distinction is not clear to me.
>
> Overfitting is a global effect of model training that is often due to training too long, whereas we argue and demonstrate that memorization is a local phenomenon that arises even when overfitting has not happened. Of course, we also expect memorization to occur when a model has been overfit, and in that sense they indeed lie on a continuum. We will clarify this in the text.
>
> > Secondly, a central assumption of the paper seems to be that memorization in generative model is undesired and problematic, but this is again not clear to me.
>
> High memorization is problematic from a privacy standpoint when training data is sensitive or confidential, but we also believe it can be an issue in general when individual observations have a large effect on the local density. Low memorization values can indeed be informative as they may indicate regions of the input space that are overrepresented in the training data. We will expand on this in our revised manuscript.
>
> > Thirdly, I think there is an important issue with the experiment described in the second paragraph of section 4.2 (related to Fig. 3). [...] (That is, by definition, the higher the likelihood under the regular model, the higher the memorization score.)
>
> We'd like to remind the reviewer that the memorization score is the log of the density ratio, and therefore it is not necessarily the case that "the higher the likelihood [...] the higher the memorization score". What the experiment demonstrates is that: (1) some outliers are always unlikely and thus not highly memorized (e.g. the image with the watermark in Fig 3a), (2) other outliers are highly memorized yet unlikely when included (e.g. the person with the large white hat), and, importantly, (3) some atypical observations are highly memorized and only likely when included (e.g. the image with pink lighting, which lies within the central 90% of log probabilities when included but has a memorization score of over 2500, making it very unlikely when excluded).
>
> > Fourthly, an important limitation of the proposed memorization score is that its absolute value is hard to interpret. [...] the authors nevertheless claim that with the VAE trained with learning rate 10e-4 “high memorization can still occur”.
>
> We will rephrase this sentence. However it is unclear to us why the reviewer is unconvinced by the provided example, as it illustrates that even though a lower learning rate leads to lower average memorization scores, we can still find observations that are highly unlikely when excluded from the training set but receive average probability when included (thus resembling the left-most blue observation in Fig 1a). If a lower learning rate would have alleviated high memorization, then we would not expect to observe this.

---

> > ### Comment · Reviewer_3dmL · 2021-08-31
> > **Response to rebuttal**
> >
> > I thank the authors for their response to my review. My apologies for responding towards the end of the discussion periods.
> >
> > Regarding the first point, I’m afraid that it is still not completely clear to me how or why memorization is fundamentally different from overfitting. The authors’ response seems to indicate that memorization is a phenomenon that starts to occur before overfitting starts to happen, but this only seems to support that these two lie on the same continuum? It is great that this will be clarified in the revised text, but this does not take away this concern. (Also because it is not indicated exactly how this will be clarified.) It seems to me that the main distinction that is claimed between memorization and overfitting is that the former is a local phenomenon and the latter is a global phenomenon. I think this is interesting. But is it really the case that overfitting is always global? Can it not be the case that overfitting is more severe in certain parts of the input space than in others?
> >
> > Regarding the third point, I’m confused by the authors’ response. Based on equation (2), it seems clear to me that if – all other things equal – the likelihood of an observation under the regular model increases, that then its memorization score also increases.
> >
> > Regarding both the third and fourth point, I think the main unaddressed issue here is that these are all just examples and individual observations. Some natural statistical variation of the memorization score is expected, and it is not evident whether the provided examples could not be explained by this natural and expected variation. I think this is particularly an issue as – as the authors acknowledge – it is not clear what constitutes a “high” memorization score.

---

> > > ### Author Response · Authors · 2021-08-31
> > > **Response to reviewer 3dmL**
> > >
> > > We thank the reviewer for the additional comments.
> > >
> > > > Regarding the first point, I’m afraid that it is still not completely clear to me how or why memorization is fundamentally different from overfitting.
> > >
> > > This concern seems to come down to definitions. Overfitting is generally understood to be a property of _model training_, i.e. a global  property of a model. This is often identified through the generalization gap and/or an increasing test error. Now, consider a hypothetical model that generalizes everywhere except for a small region of the space where the density consists of peaks on the training data. Such a model would have good test performance, but we would still say that it has memorized the observations in that small region. This could also be referred to as "local overfitting", but then it becomes a discussion of semantics.
> > >
> > > > Regarding the third point, I’m confused by the authors’ response. Based on equation (2), it seems clear to me that if – all other things equal – the likelihood of an observation under the regular model increases, that then its memorization score also increases.
> > >
> > > The reviewer did not state "all other things equal" in their original comment, hence we clarified the memorization score in our response.
> > >
> > > > Regarding both the third and fourth point, I think the main unaddressed issue here is that these are all just examples and individual observations.
> > >
> > > These are not cherry-picked examples or statistical anomalies. We can look at how many highly memorized observations are outliers (*) when they are absent from the training data and central inliers (**) when they are present in the training data. For MNIST with learning rates 1e-3 and 1e-4 there are 1161 and 635 such observations respectively, and for CelebA there are 1392 such observations.
> > >
> > > (*) Outliers: log probability when absent lower than the 5th percentile of the log probabilities of the model.
> > >
> > > (**) Inliers: within the central 90% of log prob.

---

> > > > ### Comment · Reviewer_3dmL · 2021-09-01
> > > > **Response to Rebuttal (2)**
> > > >
> > > >  *This concern seems to come down to definitions.*
> > > >
> > > > To some extent. I think it might be useful to state my original concern again, which was that there does not seem to be a clear distinction between memorization and overfitting, while this does seem to be a key claim in the paper. The authors’ comments have indicated that there does seem to be some distinction between “global overfitting” on the one hand and “local overfitting”/“memorization” on the other hand. This alleviates my original concern slightly, but I don’t think that this distinction is as fundamental as suggested/claimed in the paper.
> > > > I think this concern is supported by the observation that the difference between “local overfitting” and “memorization” only seems to come down to semantics.
> > > >
> > > >  *These are not cherry-picked examples or statistical anomalies.*
> > > >
> > > > My apologies, I should have been clearer, because this is not really what I meant. What I meant is that many of the results in section 4 are merely descriptive statistics. For example, that “more than half of the highly memorized observations fall within the central 90% of log probability values” (L218). The authors suggest that this result is interesting / unexpected / “counterintuitive”, because a considerable amount of highly memorized observations is “as likely as regular observations”.
> > > > My concern is that it is not clear what this particular value of the descriptive statistics means, and why it is different from what would be expected (by chance?). As raised in my previous comment, some natural statistical variation is always expected. Especially in the context of this paper, I think this is particularly an issue because (1) it is not clear what constitutes a “high” memorization score and (2) for the experiment related to Fig 3 there is the unaddressed confound that an increase in the likelihood under the regular model also increases the memorization score.

---

> > > > > ### Author Response · Authors · 2021-09-01
> > > > > **Response to reviewer 3dmL**
> > > > >
> > > > > We thank the reviewer for the additional comments.
> > > > >
> > > > > > This alleviates my original concern slightly, but I don’t think that this distinction is as fundamental as suggested/claimed in the paper.
> > > > >
> > > > > We don't think that we claim a fundamental distinction in our work, and as we mentioned above we agree that in some sense these concepts lie on a continuum. We will clarify this in the revised manuscript, including by writing "memorization _and_ overfitting" for the caption Fig. 1. But overfitting is almost always referred to as a global property of a model, where due to training too long or having too many parameters it learns the noise of the training data. Indeed, one definition of overfitting could be size of the generalization gap: the expected test error minus the expected training error. However we argue and demonstrate that the generalization gap itself is not a sufficiently detailed metric to assess memorization. Moreover, our memorization score allows us to assess the extent to which an _individual_ observation has been memorized, which is not possible through the generalization gap.
> > > > >
> > > > >
> > > > > > (2) for the experiment related to Fig 3 there is the unaddressed confound that an increase in the likelihood under the regular model also increases the memorization score.
> > > > >
> > > > > Again, this is not an unaddressed confounder. An increased likelihood will only given a high memorization score if the likelihood is low when the instance is not in the training data. But that is exactly what the memorization score is trying to measure. What this experiment shows is that highly memorized observations aren't simply outliers that go from extremely unlikely when absent to very unlikely when present, they can also be very unlikely when absent but just as likely as other observations when present. We indeed believe this is an interesting observation that sheds light on how probabilistic deep generative models learn from and memorize their input data.

---

### Decision · Program_Chairs · 2021-09-27

**Decision:**

Accept (Poster)

**Comment:**

The paper introduces a memorization score for generative models which quantifies how the much the log-probability of a given datapoint (under the trained model) depends on its presence in the training set, and proposes an efficient cross-validation-based estimator for it. The authors applies this metric to VAEs trained on several datasets and produce several surprising findings about memorization in such models.

This is an interesting and well written paper that sheds some light on an important problem. The approach is simple and sensible and the experimental results are quite thought-provoking. The main concerns the reviewers had about the paper were the lack of guidance about what constitutes a high memorization score and the lack of precision in the claim that memorization is different from overfitting. The authors are encouraged to address these when revising the paper, which has the potential to be an influential contribution to the field.